# An Improved Incipient Fault Diagnosis Method of Bearing Damage Based on Hierarchical Multi-Scale Reverse Dispersion Entropy

**DOI:** 10.3390/e24060770

**Published:** 2022-05-30

**Authors:** Jiaqi Xing, Jinxue Xu

**Affiliations:** Marine Electrical Engineering College, Dalian Maritime University, Dalian 116026, China; 1120190185@dlmu.edu.cn

**Keywords:** incipient fault, hierarchical multi-scale reverse dispersion entropy, feature extraction

## Abstract

The amplitudes of incipient fault signals are similar to health state signals, which increases the difficulty of incipient fault diagnosis. Multi-scale reverse dispersion entropy (MRDE) only considers difference information with low frequency range, which omits relatively obvious fault features with a higher frequency band. It decreases recognition accuracy. To defeat the shortcoming with MRDE and extract the obvious fault features of incipient faults simultaneously, an improved entropy named hierarchical multi-scale reverse dispersion entropy (HMRDE) is proposed to treat incipient fault data. Firstly, the signal is decomposed hierarchically by using the filter smoothing operator and average backward difference operator to obtain hierarchical nodes. The smoothing operator calculates the mean sample value and the average backward difference operator calculates the average deviation of sample values. The more layers, the higher the utilization rate of filter smoothing operator and average backward difference operator. Hierarchical nodes are obtained by these operators, and they can reflect the difference features in different frequency domains. Then, this difference feature is reflected with MRDE values of some hierarchical nodes more obviously. Finally, a variety of classifiers are selected to test the separability of incipient fault signals treated with HMRDE. Furthermore, the recognition accuracy of these classifiers illustrates that HMRDE can effectively deal with the problem that incipient fault signals cannot be easily recognized due to a similar amplitude dynamic.

## 1. Introduction

In the actual industrial process, a slight degree of deviation is regarded as a minor symptom. The fault with minor symptoms is defined as the incipient fault [1]. This means that the fault amplitude of incipient faults is less obvious, which increases recognition difficulty for these fault signals in the time domain or frequency domain [2].

Incipient faults are similar to each other, which is characterized by a slight deviation from the normal health condition, but each fault and normal state belong to two classes of objective existence, respectively. Therefore, it is significant to select and improve signal treatment methods to reflect the great difference.

Different from feature extraction methods, such as deep learning, which enhances the learning ability by constructing various network structures [3,4,5], the signal treatment method decreases the distinguishing difficulty of incipient fault signals by restructuring new variables which can embody more obvious difference information. The amplitude of the signal changes with the passage of time [6]. Furthermore, amplitude deviations of fault are different from those of the normal state. Many methods with the measurement of the disorder of nonlinear time series have been proposed and applied to the field of fault diagnosis [7], such as approximate entropy (AE) [8], sample entropy (SE) [9], fuzzy entropy (FE) [10], and permutation entropy (PE) [9]. For example, approximate entropy has been used in bearing fault diagnosis; different fault sizes of the bearing inner are measured by approximate entropy [11]. Compared with approximate entropy, data length does not influence the calculation of sample entropy. Sample entropy and empirical mode decomposition are combined as the battery fault detection method [12]. Fuzzy entropy introduces the idea of threshold segmentation. A Euclidean distance based multi-scale fuzzy entropy method has been proposed to diagnose bearing faults, which measures the similarity of two vectors with continuous values from zero to one based on the Euclidean distance of the two vectors [13]. An improved FE named refined composite multi-scale fuzzy entropy (RCMFE) has been applied to diagnose the significant bearing fault [14]. Being different from AE, SE, RCMFE, and FE, PE compares and analyzes the order of amplitude values to obtain the corresponding feature information rather than considering the value of the time series. Therefore, PE possesses the merit of fast computation. However, it ignores the difference between different amplitude values, which will cause the omission of important amplitude information. A method based on variational mode decomposition and permutation entropy has been used in wind turbine roller bearing fault diagnosis, and its feature extraction ability is superior to PE [15]. All the same, PE and its improved methods play an important role in fault diagnosis, such as weighted PE (WPE) [16], dispersion entropy (DE) [17], reverse permutation entropy (RPE) [18], and reverse dispersion entropy (RDE) [19]. For example, WPE has been combined with an improved support vector machine as a bearing fault classification method [20]. Both WPE and DE add amplitude information to PE, but DE is proposed to generate different fluctuation dispersion patterns by mapping each element of a measured series to different classes, which means that DE has faster calculation and the signals that are treated with DE have better separability [17,21]. To promote the feature extraction ability of DE, an improved refined composite multi-scale dispersion entropy (RCMDE) has been proposed to isolate bearing fault data provided by Case Western Reserve University [22]. The optimized method RPE is defined as the distance from white noise and it is better than PE in feature extraction [18]. The merits of DE and RPE are combined in RDE; therefore, RDE has better feature extraction ability than DE and RPE [19,23]. Based on RDE, multi-scale reverse dispersion entropy (MRDE) [24] has been proposed in 2022; it can describe the disorder of the signal from different scales, which solves the problem that RDE ignores useful information on other scales, and it obtains better performance on feature extraction of the ship-radiated noise.

So far, a lot of recent work has focused on regular fault data for testing the optimized diagnosis method and proving the promotion of recognition accuracy. Different from regular faults, once incipient fault occurs in a system, its amplitude difference from the normal state is more slight [1]. Regular methods, which only extract amplitude change information, may not be satisfied to deal with incipient fault data. Compared with normal state signals, incipient fault signals are in a higher frequency band. Because RDE and MRDE only extract amplitude difference features in the low frequency band, the obvious incipient fault feature with higher frequency range will be omitted, which will cause the lower incipient fault recognition accuracy. To overcome the defect of MRDE, an improved hierarchical multi-scale reserve dispersion entropy (HMRDE) method is proposed to enhance the separability of signals.

The contributions are summarized as follows:(1)A new fault extraction approach, named HMRDE, based on MRDE is proposed to extract obvious difference features with various frequency ranges. It introduces hierarchical thought to MRDE and uses hierarchical nodes to analyze the frequency difference features of incipient fault signals for the first time.(2)HMRDE enhances the disorder difference of each state by calculating the change deviation with a high-frequency operator and reflects this difference by entropy values of hierarchical nodes obviously, which helps classifiers greatly in recognizing incipient faults.

The remainder is organized as follows. Section 2 briefly describes the motivation of the proposed method for incipient faults and describes the proposed method, HMRDE. Section 3 gives a numerical example to test the feature extraction ability of HMRDE for similar signals and a real fault diagnosis experiment to test the effectiveness of HMRDE for real incipient faults. The findings and their implications are discussed in Section 4. Finally, conclusions are drawn in Section 5.

## 2. Aim Formulation and Methods

### 2.1. Aim Formulation

Fault signals and normal signals are two kinds of objective existence. Both fault signals and normal signals have inherent center frequency. Compared with normal signals, fault signals are in a higher frequency band. Different kinds of faults have different frequency domain characteristics. Normal signals and fault signals with fixed center frequency can be analyzed in the frequency domain. Each health status signal can be expressed in the form of a periodic f(t) with a period *T*, (*T* can approach positive infinity). Fourier decomposition of f(t) can be defined as
(1)$$f\left(t\right)=d+\sum_{n=1}^{\infty }\left(a_{n}\cos\left(\frac{2\pi n}{T}t\right)+b_{n}\sin\left(\frac{2\pi n}{T}t\right)\right)$$
where *d* represents constant term, $a_{n}$ and $b_{n}$ denote amplitudes of periodic function $\sin(w_{n}t)$ and $\cos(w_{n}t)$ with frequency $w_{n}$, $w_{n}=\frac{2\pi n}{T}$. In the time domain, the amplitude difference of each fault is not obvious. For example, assume a normal state and one incipient fault, $f_{0}\left(t\right)$ and $f_{1}\left(t\right)$, respectively, is described through sin(wt) as
(2)$$\left\{\begin{array}{c} f_{0}\left(t\right)=a_{1}\sin\left(w_{1}t\right)\\ f_{1}\left(t\right)=a_{1}\sin\left(w_{1}t\right)+\sigma \sin\left(w_{2}t\right) \end{array}\right.$$
where $1\leq w_{1}<w_{2}$ and σ is set to be $0.02a_{1}$ [25]. Thus, their first order derivatives can be calculated as
(3)$$\left\{\begin{array}{c} f_{0}^{\prime }\left(t\right)=a_{1}w_{1}\cos\left(w_{1}t\right)\\ f_{1}^{\prime }\left(t\right)=a_{1}w_{1}\cos\left(w_{1}t\right)+\sigma w_{2}\cos\left(w_{2}t\right) \end{array}\right.$$

It can be seen that $|\left(f_{1}^{\prime }\left(t\right)-f_{0}^{\prime }\left(t\right)\right)\left|\geq \right|\left(f_{1}\left(t\right)-f_{0}\left(t\right)\right)|$, which shows that the amplitude difference between $f_{1}^{\prime }$ and $f_{0}^{\prime }$ is greater than that between $f_{1}$ and $f_{0}$. $f_{1}^{\prime }$ is more obviously different from $f_{0}^{\prime }$.

This shows that the natural frequency characteristics of the incipient fault signal are obviously different from those of the normal signal, and the natural center frequency of the incipient fault is in a higher frequency band.

Thus, the motivation of the proposed method regarding the recognition of incipient faults is that the signal treatment method needs to consider the obvious differences of each incipient fault from others in higher frequency ranges, and it reflects them greatly.

### 2.2. Methods

Hierarchical multi-scale reverse dispersion entropy defeats the defect that multi-scale reverse dispersion entropy only analyzes the obvious differences of each incipient fault from others in low frequency ranges.

For a time series {x(1),x(2),⋯,x(n)}, we define the averaging operator $Q_{0}$ and high-frequency operator $Q_{1}$ as follows [26]
(4)$$Q_{0}\left(x\right)=\frac{x(i)+x(i+1)}{2},\;i=1,2,\cdots ,n$$
(5)$$Q_{1}\left(x\right)=\frac{x(i)-x(i+1)}{2},\;i=1,2,\cdots ,n$$
where $Q_{0}\left(x\right)$ and $Q_{1}\left(x\right)$ can be regarded as approximations of a filtering smooth operation and an average backward differential operation, and they can depict the low frequency and high frequency information of the time series respectively.

The matrix form of operators $Q_{j}^{k}$ (j=0,1) at hierarchical layer *k* can be expressed as
(6)$$Q_{j}^{k}={\left[\begin{array}{cccccccc} \frac{1}{2} & \underset{2^{k-1}-1}{\underset{︸}{0\cdots 0}} & \frac{{\left(-1\right)}^{j}}{2} & 0 & \cdots & 0 & 0 & 0\\ 0 & \frac{1}{2} & \underset{2^{k-1}-1}{\underset{︸}{0\cdots 0}} & \frac{{\left(-1\right)}^{j}}{2} & \cdots & 0 & 0 & 0\\ \cdots & \cdots & \cdots & \cdots & \cdots & \cdots & \cdots & \cdots \\ 0 & 0 & 0 & 0 & 0 & \frac{1}{2} & \underset{2^{k-1}-1}{\underset{︸}{0\cdots 0}} & \frac{{\left(-1\right)}^{j}}{2} \end{array}\right]}_{a\times b}$$
where $a=n-2^{k}+1$ and $b=n-2^{k-1}+1$. The hierarchical decomposition structure is exposed in Figure 1.

Furthermore, hierarchical nodes can be calculated by
(7)$$X_{k,e}=Q_{r_{k}}^{k}\cdot Q_{r_{k-1}}^{k-1}\cdots \cdots \cdot Q_{r_{1}}^{1}\cdot X$$
where X={x(1),x(2),⋯,x(n)}, and vector $\left[r_{1},r_{2},\cdots ,r_{k}\right]$ is given by non-negative integer *e*
(8)$$e=\sum_{m=1}^{k}2^{k-m}r_{m}$$
where $e\in \{0,1,\cdots ,2^{k}-1\}$, $r_{m}$ is 0 or 1, which denotes the average or difference operator at layer *m*. The partial calculation process is shown in Figure 2.

In Figure 1 and Figure 2, the larger the value of *k*, the higher the utilization rate of high-frequency operators. The higher frequency range of time series is analyzed by the node on the right side of HMRDE. In a certain unique frequency band, the change of one incipient fault signal must be obviously different from other faults. Therefore, the difference information from low frequency range to high frequency range can be analyzed by increasing layer *k* suitably.

Entropy is a reflection of signal disorder, so this signal difference can be measured by multi-scale reverse dispersion entropy. For a certain component with length $n-2^{k}+1$, $X_{k,e}=\left\{x_{k,e}\left(1\right),x_{k,e}\left(2\right),\cdots ,x_{k,e}\left(n-2^{k}+1\right)\right\}$, the coarse-grained result is as follows
(9)$$x_{k,e}^{s}\left(j\right)=\frac{1}{s}\sum_{i=(j-1)s+1}^{js}x_{k,e}\left(i\right)$$
where *s* is the scale factor of MRDE. Map $X_{k,e}^{s}=\left\{x_{k,e}^{s}\left(1\right),x_{k,e}^{s}\left(2\right),\cdots ,x_{k,e}^{s}\left(\left(n-2^{k}+1\right)/s\right)\right\}$ to $Y_{k,e}^{s}$ using the normal cumulative distribution function, which is expressed as
(10)$$y_{k,e}^{s}\left(j\right)=\frac{1}{\sigma \sqrt{2\pi }}\int_{-\infty }^{x_{k,e}^{s}\left(j\right)}e^{-\frac{{\left(t-\mu \right)}^{2}}{2\sigma^{2}}dt}$$
where μ and $\sigma^{2}$ denote expectation and variance, respectively, and $y_{k,e}^{s}\left(j\right)$ ranges from 0 to 1. Then, map each $y_{k,e}^{s}\left(j\right)$ to the sequence {1,2,⋯,c} by linear transformation as follows
(11)$$z_{k,e}^{s,c}\left(j\right)=round\left(c*y_{k,e}^{s}\left(j\right)+0.5\right)$$
where round(·) represents the integral function and *c* is the class number. This formula limits the magnitude of $y_{k,e}^{s}\left(j\right)$ to an integer range of [1,c]. The embedding vector of reconstructed matrix $Z_{k,e}^{s,c,m}=\left\{z_{k,e}^{s,c,m}\left(1\right),z_{k,e}^{s,c,m}\left(2\right),\cdots ,z_{k,e}^{s,c,m}\left(\left(n-2^{k}+1\right)/s-\left(m-1\right)\tau \right)\right\}$ with the embedding dimension *m* is defined by
(12)$$z_{k,e}^{s,c,m}\left(j\right)=\left[z_{k,e}^{s,c}\left(j\right),z_{k,e}^{s,c}\left(j+\tau \right),\cdots ,z_{k,e}^{s,c}\left(j+\left(m-1\right)\tau \right)\right]$$
where τ represents the time delay. Each $z_{k,e}^{s,c,m}\left(j\right)$ corresponds to a dispersion mode which can be described by $\left[\pi_{v_{0},\cdots ,v_{m-1}}\right]$. Calculate the relative frequency of each dispersion mode by the following equation
(13)$$p_{j}\left(\pi_{v_{0},\cdots ,v_{m-1}}\right)=\frac{Number(\pi_{v_{0},\cdots ,v_{m-1}})}{\left(\left(n-2^{k}+1\right)/s-\left(m-1\right)\tau \right)}$$
where Number(·) is the number of mappings from $z_{k,e}^{s,c,m}\left(j\right)$ to $\left\{\pi_{v_{0},\cdots ,v_{m-1}}\right\}$. Reverse dispersion entropy (RDE) is used to calculate the entropy value of each node $X_{k,e}$ in the hierarchical layer. RDE is defined as the distance to white noise by combining distance information [19]. The entropy value of each node $X_{k,e}^{s}$ with scale factor *s* in the hierarchical layer can be expressed as [19]
(14)$$RDE\left(X_{k,e}^{s}\right)=\sum_{j=1}^{c^{m}}{\left(p_{j}-\frac{1}{c^{m}}\right)}^{2}$$
when $p_{j}=\frac{1}{c^{m}}$, the value of $RDE(X_{k,e}^{s})$ is 0 (minimum value) [19]. This means that the smaller the RDE value is, the more disorderly the signal is. The HMRDE of a given time series *X* is defined as
(15)$$HMRDE\left(X\right)=[RDE\left(X_{k,0}^{s}\right),RDE\left(X_{k,1}^{s}\right),\cdots ,RDE\left(X_{k,2^{k}-1}^{s}\right)]$$

Notably, $X_{k,0}$ is generated by *k* operations of filtering smooth and $X_{k,2^{k}-1}$ is acquired through *k* calculations of mean change deviation of adjacent sample values. $X_{k,0}$ equals sample entropy at $2^{k}$ scale in multi-scale analysis. Based on HMRDE, the proposed fault diagnosis scheme for rolling bearings is given in Figure 3. The specific steps for the proposed scheme are given as follows.

Step 1: Collect vibration signals with *l* classes. Each type of data file has the same number of time series samples, and each series sample has the same number of consecutive non-overlapping points. Divide the signals randomly into two groups: one for the training samples, which can be used to optimize the parameters of the method, and the other for the testing samples.

Step 2: Determine the hyperparameter adjustment range and set the hyperparameter initialization. For example, the adjustment range of layer *k* is $\left\{n,n+1,\cdots ,n_{max}\right\}$.

Step 3: Select optimal hyperparameters of HMRDE. In the training stage, the hyperparameters are adjusted, and the same classifier is used to test the effectiveness of different parameter setting methods. Select the HMRDE parameter setting with the best feature extraction effect. Under this parameter setting, the data processed by HMRDE has better distinguishability, and the same classifier can achieve higher classification accuracy. The flowchart of hyperparameter optimization of layer *k* is shown in Figure 4. Assume that the optimal hyperparameter layer *k* is *m*.

Step 4: Hierarchical decomposition of testing signals using HMRDE with optimal hyperparameters, which generates hierarchical nodes of layer *k* (k=m). Then, calculate entropy values of these nodes as the fault feature vectors.

Step 5: Use the classifier to classify the test dataset processed by HMRDE.

## 3. Results

### 3.1. Case 1: Numeral Example

Assume the normal condition $f_{0}\left(t\right)$ and incipient fault signals $f_{1}\left(t\right)$ are described as
(16)$$\left\{\begin{array}{c} f_{0}\left(t\right)=\sin\left(50t\right)\\ f_{1}\left(t\right)=\sin\left(50t\right)+0.02\sin\left(100t\right) \end{array}\right.$$

The amplitude of incipient faults is similar to that of the health condition from Equation (Equation 16). Figure 5 shows that there must be relatively obvious fault features in the higher frequency range when the amplitude difference information with the low frequency range is very hidden. The first order derivative of the time series under two health conditions is calculated as
(17)$$\left\{\begin{array}{c} {f_{0}}^{\prime }\left(t\right)=50*\sin\left(50t\right)\\ {f_{1}}^{\prime }\left(t\right)=50*\sin\left(50t\right)+0.02*100*\sin\left(100t\right) \end{array}\right.$$
and it is depicted in Figure 6. The difference of $f_{1}\left(t\right)$ from $f_{0}\left(t\right)$ shown in Figure 6 is more obvious than that depicted in Equation (Equation 16), which indicates that relatively obvious difference information exists in the higher frequency range rather than in the low frequency band. Figure 6 shows that the standardized derivative values of $f_{1}^{\prime }$ are more obviously different from those of $f_{0}^{\prime }$, which illustrates that difference information with a higher frequency band can be reflected by a derivative operation. At the same time, the relative obvious fault features with higher frequency range also can be reflected by a high-frequency operator, as shown in Figure 7.

The node entropy values of time series under two health conditions are depicted in Figure 7, and these node entropy values are calculated by HMRDE, where layer *k* is 2, embedding dimension *m* is 3, time delay τ is 1, scale factor *s* is 1, and class number *c* is 5. In Figure 7, the entropy value of $X_{2,3}$ of $f_{1}$ is lower than that of $f_{0}$, which is easily distinguished. The disorder of the signal treated with a high-frequency operator can be effectively reflected by the entropy values of the high frequency node.

### 3.2. Case 2: Dataset Provided by Padborn University in Germany

In order to verify the practicability of HMRDE, the dataset provided by Padborn University in Germany [25,27] is used to carry out the real incipient fault diagnosis experiment. Specifically, the fault data generated by the accelerated lifetime test was used in the recognition of incipient fault in 2020 [28].

The basic setup of operation parameters is that N = 1500 rpm, M = 0.7 Nm, and F = 1000 N [25]. Then, fault data are assigned to five levels according to Table 1.

The dataset consists of three kinds of health conditions: normal, inner ring (IR) fault, and outer ring (OR) fault; the types of these faults are: single point (S) fault, repetitive (R) fault, and multiple (M) fault. All the incipient faults of rolling bearing belong to level 1 (extent of damage: 0–2%). A detailed description of the datasets is illustrated in Table 2.

The number of sample values is 256,000 for each fault and the length of each time series input is 3000. There are 85 time series inputs. Then, 60% of the time series inputs is randomly chosen for training and the remaining 40% is chosen for testing.

The waveform of the two random time series samples under five bearing conditions is sketched in Figure 8. It indicates that the amplitudes of five health condition signals are similar.

Then, conclude the feature frequency spectrum using FFT transform, as shown in Figure 9. The sample frequency is 64 kHz and the sample length is 256,000. In Figure 9, it is difficult to distinguish the five health conditions through amplitudes; although, the amplitudes with a low frequency range are high. However, the frequency features of five health conditions are obviously different from each other in the frequency band marked by the red, five pointed star, and the frequency spectrum in this frequency range is sketched in Figure 10. It illustrates that there must be obvious frequency difference information of the five health condition signals in a certain unique frequency range; although, their amplitudes are very low and similar. The frequency bands with distinct fault characteristics for A01, B01, I01, I02, and N01 are described as Fre.(A01), Fre.(B01), Fre.(I01), Fre.(I02), and Fre.(N01).

In the real application of HMRDE, there are five parameters that need to be determined. Because the low-frequency smoothing operation in HMRDE can be regarded as an MRDE calculation with scale, for example, entropy values of $X_{k,0}$ of HMRDE are equal to MRDE values of *X* with scale $2^{k}$, the scale of HMRDE should not be larger. Because the obvious difference information exists in 2690∼3820 Hz from these two figures, layer *k* cannot be selected too large; usually, it is set as 2–6. The embedding dimension *m* and the number of classes *c* can be 3 and 5, respectively, and the time delay τ is 1. For more information about the parameters *m*, *c*, and τ, please refer to the literature [22,24]. Assume the embedding dimension *m* is 3, the time delay τ is 1, the scale factor *s* is 1, the layer *k* is 3, and the class number *c* is 5. Then, calculate the HMRDE of the five health condition data. The node entropy values of a time series under five bearing conditions are shown in Figure 11. In Figure 11, node $X_{3,5}$ and $X_{3,7}$ of five health conditions are more easily distinguished than node $X_{3,0}$, $X_{3,2}$, $X_{3,4}$, and $X_{3,6}$, which explains that the obvious difference information of the dataset exists in some unique higher frequency ranges rather than low frequency ranges.

A01 with lower frequency range Fre.(A01) is more ordered than other health conditions, and the entropy values of nodes of A01 are larger than others, as shown in Figure 11. The difference of the disorder of each fault is more easily separated through calculations of mean change deviation with high-frequency operators.

At the same time, in the test of incipient fault recognition, the setting of the parameters of the proposed method is important. To test the advantage of the proposed method, the different classifiers are selected to recognize the incipient faults. To guarantee the reliability of the experiment, eight classifiers are selected to test the effectiveness of HMRDE. The selected classifiers are linear discriminant (LD), linear support vector machine (SVM), medium Gaussian support vector machine (MGSVM), quadratic support vector machine (QSVM), coarse K nearest neighbors (CKNN), bagged trees (BT), medium tree (MT), and boosted trees (BoT).

Experiments for each setting are repeated five times. The influence of the selected layer *k* on recognition accuracy of classifiers is shown in Figure 12, which depicts the highest recognition accuracy of eight classifiers in the training phase. In the training phase, when k=2,3,4,5,6, the incipient fault recognition accuracy is on average 78.3±0.69%, 93.9±0.36%, 94.1±1.17%, 96.5±0.99%, and 94.6±1.11%. Therefore, the layer *k* of HMRDE can be 5 for these five health condition signals.

Figure 13 shows the recognition accuracy with different layers in the testing phase; it can be seen that the highest accuracy is on average 97.7±0.83% when k=5. Furthermore, Figure 14 displays confusion matrix results of SVM for the inputs treated with the proposed method. It shows that the data distinguished by simple classifier SVM is more easily recognized after being treated with HMRDE, and the difference information of incipient fault inputs is reflected greatly with HMRDE. Bearing data with different conditions are described in Table 3 [25]. To test the effectiveness of HMRDE for data under different conditions, the settings of HMRDE are the same in these tests, and experiments for data under each condition are repeated five times. In these experiments, the settings of HMRDE are m=3, τ=1, s=1, k=5, and c=5. Effectiveness test results of HMRDE for data with different conditions are shown in Figure 15, which depicts the highest recognition accuracy of eight classifiers. In Figure 15, the highest recognition accuracies of eight classifiers with data treated with HMRDE are 97.65±0.83%, 91.38±1.12%, 95.8±1.28%, and 91.2±0.65%, which are all higher than 90%. This illustrates that HMRDE can effectively extract incipient fault features from incipient data under different conditions.

Assume the inputs treated with HMRDE, MRDE, and the standardization method are named ’HMRDE data’, ’MRDE data’, and ’Stand. data’, respectively. Here, standardization method refers to the zero-mean normalization method. The classification accuracy of these classifiers for ’HMRDE data’, ’MRDE data’, and ’Stand. data’ is summarized in Figure 16. Experiments for data with different treatments are repeated five times. In Figure 16, the best average accuracy of the selected classifier for ’Stand. data’ is 85.9% and that for ’HMRDE data’ is 97.7%. Compared with ’Stand. data’, the accuracy of all classifiers for data treated with HMRDE is increased by 11.8%, 17.6%, 77.7%, 63.5%, 76.6%, 64.1%, 65.3%, and 48.9%, and it is increased by 79.5%, 74.1%, 69.5%, 72.9%, 70.0%, 69.5%, and 53% compared to ’MRDE data’, whose classifier classification accuracy is 21.1%.

## 4. Discussion

Fault state and normal state are two kinds of objective existence, but MRDE cannot extract relatively obvious difference successfully, which decreases the recognition accuracy, as depicted in Figure 16. Once incipient fault occurs in a system, the values of samples slightly fluctuate in the time domain, but amplitude dynamic deviation speed and frequency change may be obvious, as shown in Equation (Equation 17) and Figure 6. The obvious difference information might exist in a higher frequency band, as shown in Figure 5 and Figure 10, and the difference information with the high frequency band can be extracted by calculating the mean change deviation of sample values, such as derivative operation and high-frequency operator filtering, which is manifested in Equation (Equation 17), Figure 6, Figure 7 and Figure 11. Furthermore, the difference in terms of disorder of each health condition is obviously reflected by MRDE values of hierarchical nodes, which enhances the separability of fault features and promotes the recognition accuracy of various classifiers, as depicted in Figure 7, Figure 11, and Figure 16. It takes between 2.5 and 3.0 s to compute HMRDE at five layers for a time series with 3000 points. HMRDE increases the calculation complexity compared to MRDE, and it has the same shortcoming that its hyperparameter selection requires expert experience. However, HMRDE defeats the drawback that MRDE omits frequency change features, and HMRDE compares favorably with deep learning approaches which require more hyperparameter adjustments and a more complex learning process.

In the practical application of the fault identification method of rolling bearing, the longer the transmission pathway of the fault signal, the greater the interference of the signal, and the less obvious the periodic pulse under the influence of noise. In order to filter out the noise of higher frequency band and effectively identify the fault signal with the long pathway, the generated hierarchical nodes can be low-pass filtered by increasing the scale value of HMRDE, and the higher frequency noise existing in the nodes can be filtered out, so that HMRDE can handle the fault signal with a longer pathway. Other denoising methods, such as wavelet denoising and empirical mode decomposition, etc., can also be used for signal pre-denoising. However, it may increase the computational complexity and the time of diagnostic methods. Furthermore, in the practical application of the method, after the fault diagnosis model is trained with data from a single condition, the optimized diagnosis model needs to be extended to other operating conditions. Figure 15 shows that HMRDE with the same hyperparameter setting has good feature extraction ability for data in different environments.

So far, a lot of work has focused on regular fault feature extraction through various entropy methods [29,30,31] but how to optimize entropy methods to extract incipient fault features is still in the early phase. Therefore, the improvement of entropy methods to extract incipient fault features can be regarded as the future research direction; this research direction needs to consider characteristics of incipient fault signals to overcome problems of entropy methods in incipient fault sample processing.

## 5. Conclusions

To solve the problem that it is difficult to extract fault features from incipient fault signals, an improved HMRD method is proposed based on MRDE. The filter smoothing operator and average backward deviation operator are used to extract the relatively obvious difference information between incipient fault signals in different frequency ranges and normal signals. By selecting the appropriate number of layers, the samples are smoothed and backwardly differentiated in different degrees, and the hierarchical nodes which can reflect the difference features in different frequency domains are obtained. Entropy values of hierarchical nodes are calculated by MRDE, and these entropy values are taken as new characteristic variables. It enhanced the disorder difference of each state signal and the distinguishing ability of classifier inputs, which solves the problem that MRDE omits obvious fault features in a higher frequency range and gives classifiers a higher classification accuracy. The use of HMRDE features for incipient fault classification has been been introduced and its effectiveness is verified with the use of a numeral example and a dataset generated by accelerated lifetime tests. The incipient fault recognition accuracy of LD, SVM, MGSVM, QSVM, CKNN, BT, MT, and BoT for the input treated with HMRDE is much higher than that for the data treated with MRDE and normalization processing, and HMRDE does not need to consume lots of time. Furthermore, for incipient data under different conditions, effectiveness test results of HMRDE with the same hyperparameter settings are excellent. These depict the effectiveness of HMRDE in incipient fault feature extraction.

## Figures and Tables

**Figure 1 entropy-24-00770-f001:**
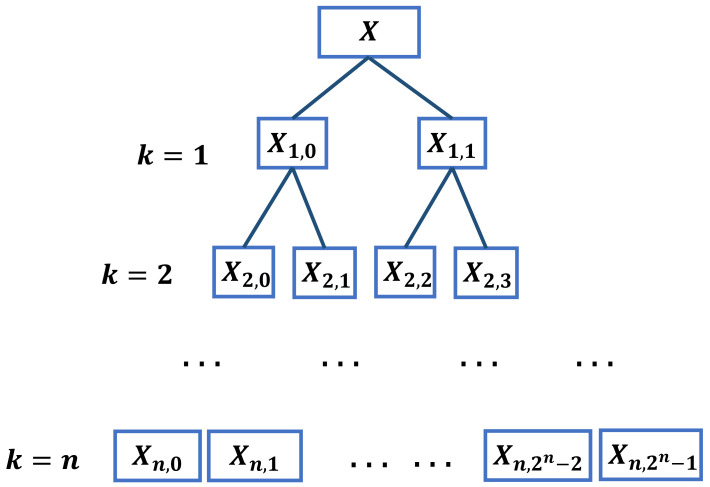
The hierarchical decomposition structure.

**Figure 2 entropy-24-00770-f002:**
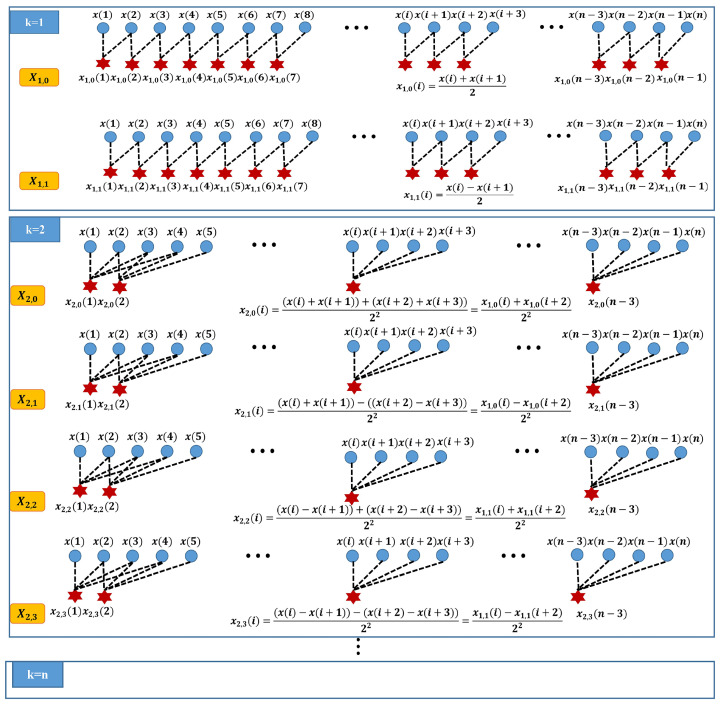
The partial calculation process regarding hierarchical decomposition structure.

**Figure 3 entropy-24-00770-f003:**
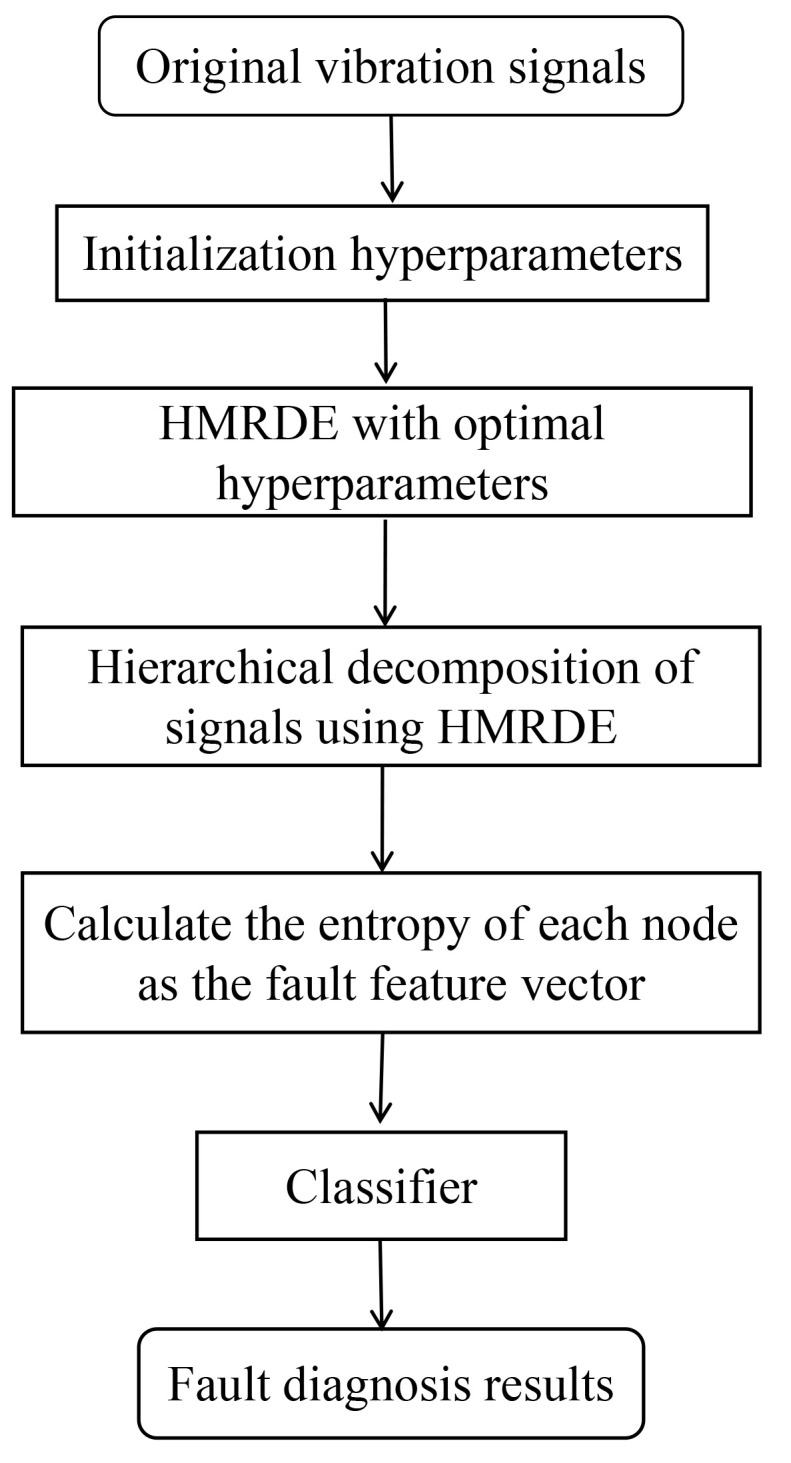
The proposed fault diagnosis scheme for rolling bearings.

**Figure 4 entropy-24-00770-f004:**
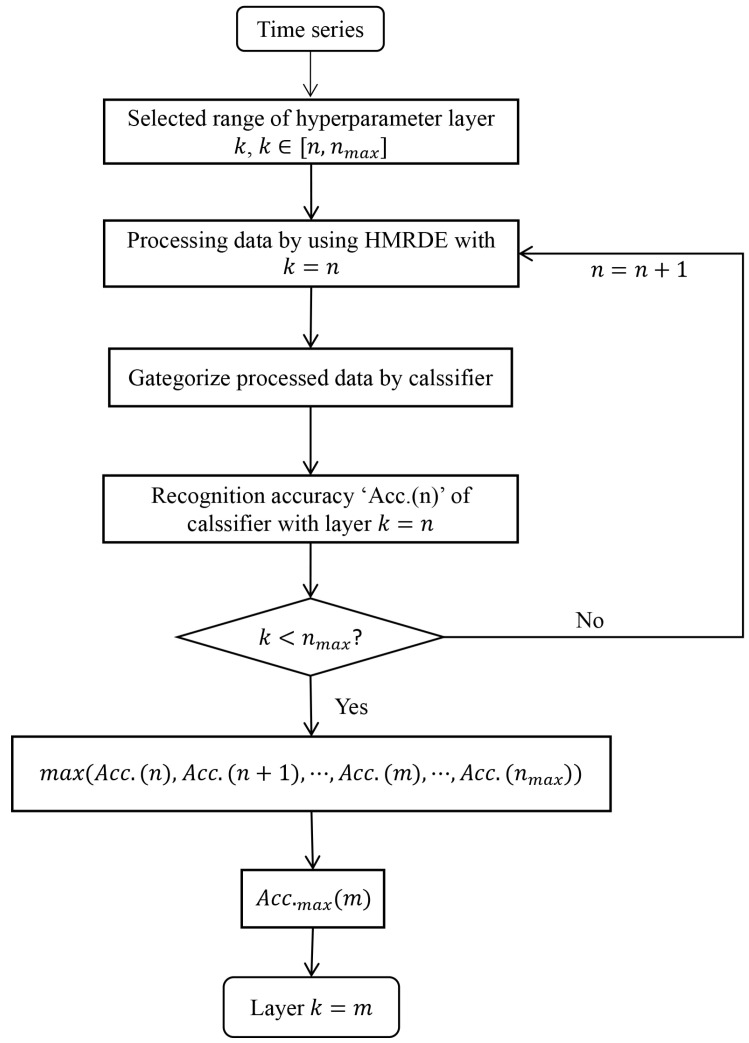
The flowchart of hyperparameter optimization of layer *k*.

**Figure 5 entropy-24-00770-f005:**
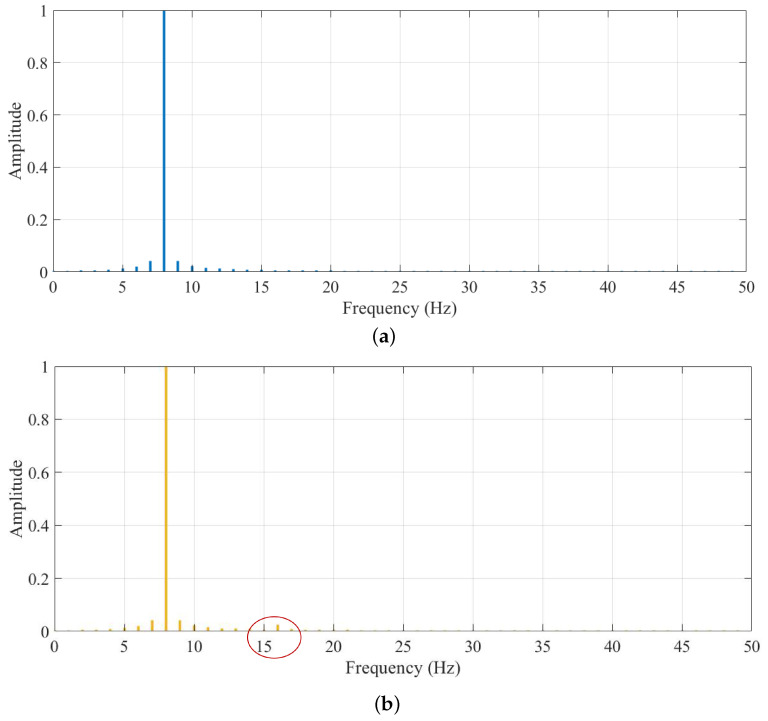
The frequency spectrum of time series under two health conditions. (**a**) Frequency spectrum of $f_{0}$; (**b**) Frequency spectrum of $f_{1}$.

**Figure 6 entropy-24-00770-f006:**
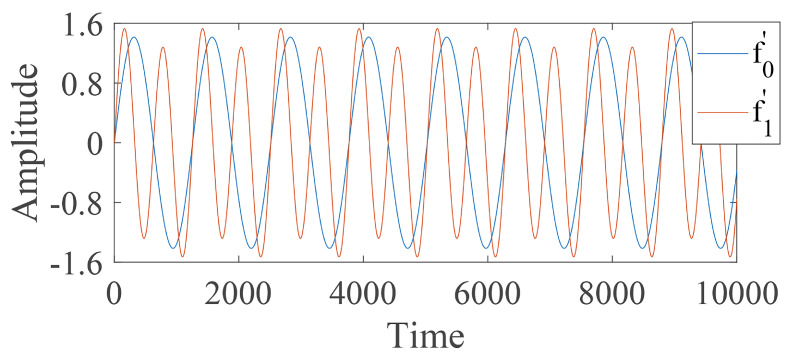
The waveform of standardized first order derivative values of time series under two health conditions.

**Figure 7 entropy-24-00770-f007:**
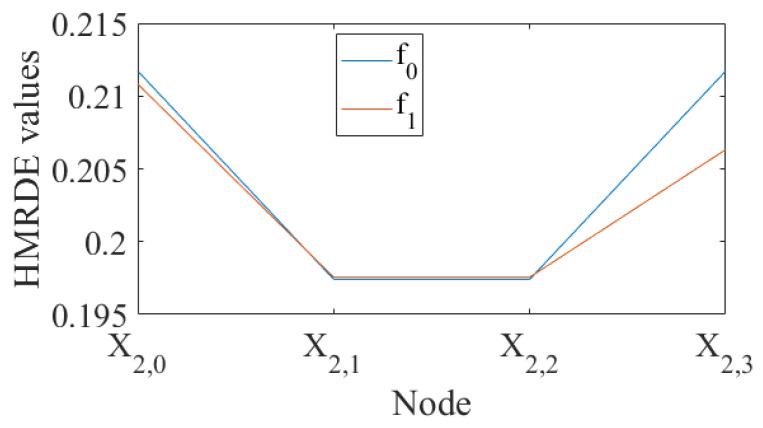
The node entropy values of time series under two health conditions.

**Figure 8 entropy-24-00770-f008:**
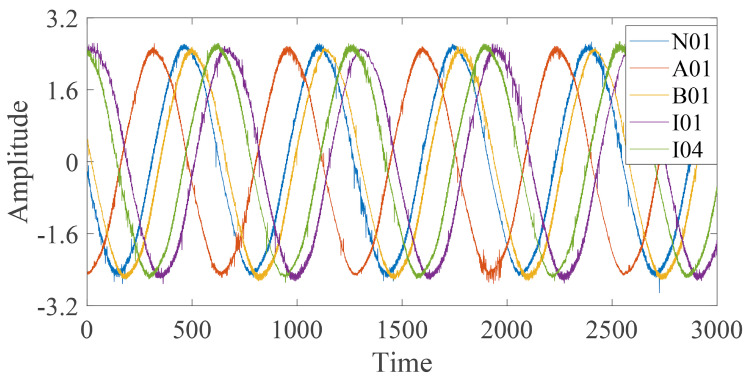
Waveform of time series under five bearing health conditions.

**Figure 9 entropy-24-00770-f009:**
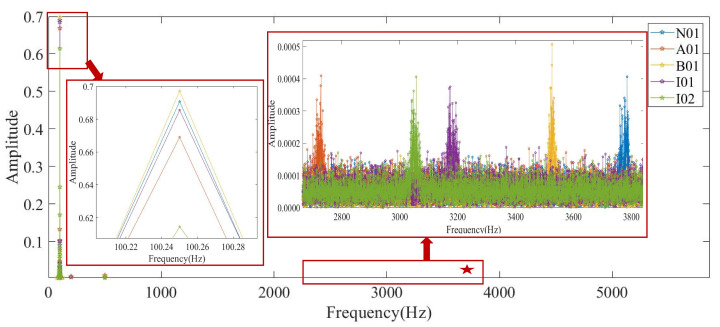
The frequency spectrum of bearing samples.

**Figure 10 entropy-24-00770-f010:**
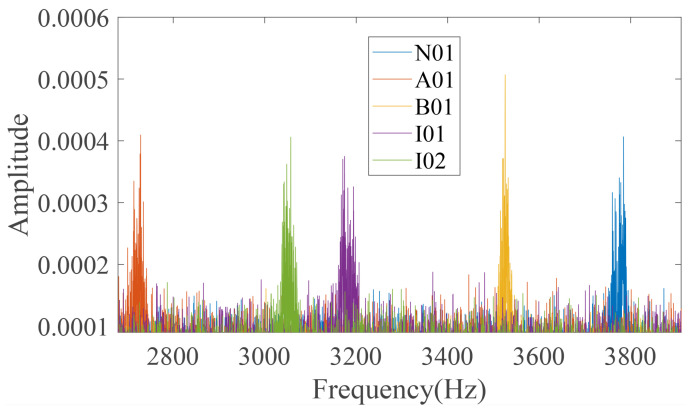
The frequency spectrum part marked by red, five pointed star.

**Figure 11 entropy-24-00770-f011:**
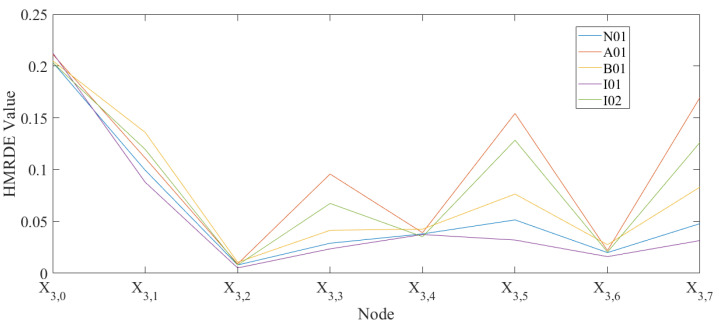
The node entropy values of a time series under five bearing conditions.

**Figure 12 entropy-24-00770-f012:**
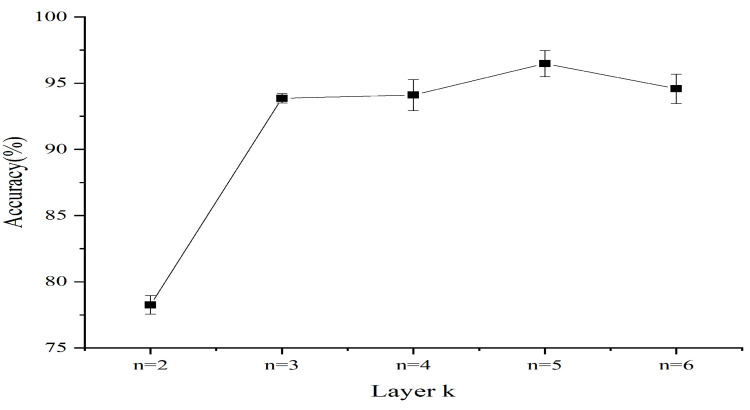
Recognition accuracy with different layers in training phase.

**Figure 13 entropy-24-00770-f013:**
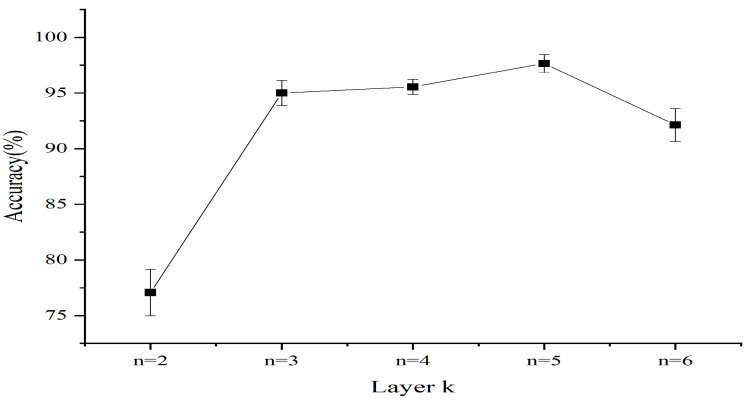
Recognition accuracy with different layers in testing phase.

**Figure 14 entropy-24-00770-f014:**
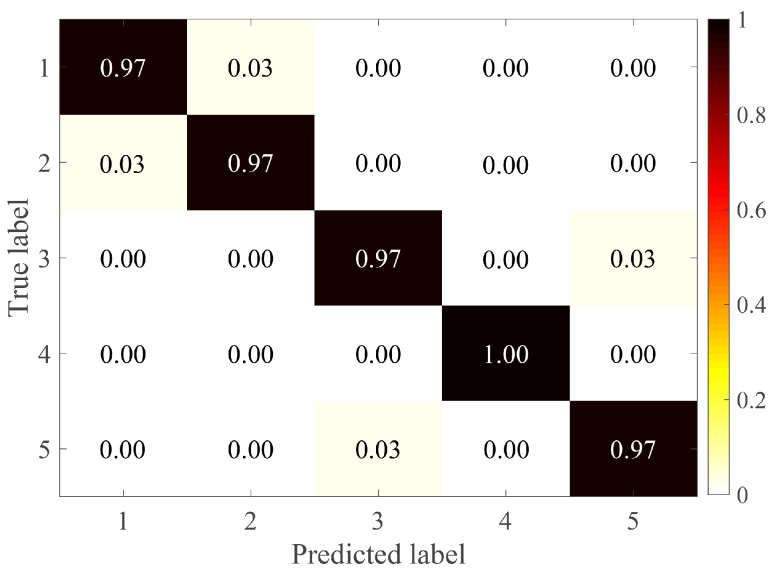
Classification results of SVM for the inputs treated with the proposed method.

**Figure 15 entropy-24-00770-f015:**
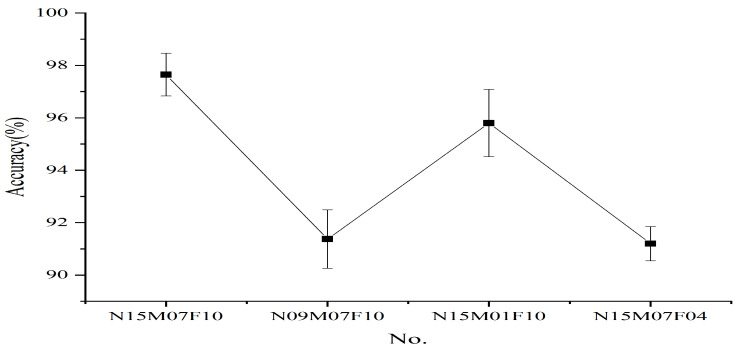
Effectiveness test results of HMRDE for data under different conditions.

**Figure 16 entropy-24-00770-f016:**
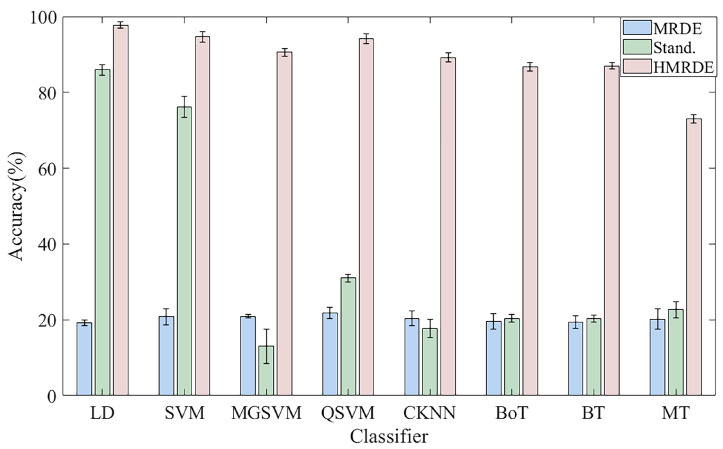
Classification accuracy of different classifiers for the data treated with HMRDE and standardization method.

**Table 1 entropy-24-00770-t001:** Damage levels to determine the extent of damage.

Damage Level	Assigned Percentage	Limits for Bearing
1	0–2%	≤2 mm
2	2–5%	>2 mm
3	5–15%	>4.5 mm
4	15–35%	>13.5 mm
5	>35%	>31.5 mm

**Table 2 entropy-24-00770-t002:** Detailed description of datasets.

Code *n*	Component *m*	Combination	Characteristic	Level
N01	–	–	–	–
A01	OR	R	distributed	1
B01	OR+IR	M	distributed	1
I01	IR	M	single point	1
I02	IR	R	single point	1

**Table 3 entropy-24-00770-t003:** Bearing data with different conditions.

Condition Number	Rotational Speed (rpm)	Load Torque (Nm)	Radial Force (N)	Name of Setting
1	1500	0.7	1000	N15M07F10
2	900	0.7	1000	N09M07F10
3	1500	0.1	1000	N15M01F10
4	1500	0.7	400	N15M07F04

## Data Availability

The data presented in this study are openly available on the KAt-DataCenter website of the Chair of Design and Drive Technology, Paderborn University, Germany: http://mb.uni-paderborn.de/kat/datacenter (accessed on 12 December 2021).

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
