# Peer review of "An Improved Incipient Fault Diagnosis Method of Bearing Damage Based on Hierarchical Multi-Scale Reverse Dispersion Entropy"

_entropy, 2022, doi:10.3390/e24060770_

Round 1

Reviewer 1 Report

This study was conducted about multi-scale reverse dispersion  entropy (MRDE). It considers difference information with low frequency range through filtering  smooth operation, which will omit relatively obvious deviation difference features with higher  frequency band and get worse recognition accuracy. There was evaluated conventional typical data treatments.

The new configuration Figure could be enabled that are created with many layers and optimum conditions should be shown by scheme.

Check space and caption mistakes in MS.

Multi-scale  reverse dispersion entropy cannot analyse the obvious differences of each incipient fault  from others in multiple frequency ranges, why?

- it seems process did not remove difference in incipient fault  effectively, why?

Process is not well-recognised alternative resource recoverful applications despite its inherent advantages and robustness. The optimum conditions for the process need to be shown. The results showed the modelling was well developed under various conditions, which was a good results. The work states that among the other studies, current study type was novel, in what sense?

Give long expressions of the abbreviations.

Direct aims section needed.

Figures should be sharper.

Fig. 1, need clearer structure and error estimations and showing the meanings of parameters.

Process related should be studied in order to evaluate the real applicability,  operation and performances for operation optimisation while meeting and undercutting requirements. 

Some of the model pathways may be questionable to apply in real-scale due to difficulties and, elaborate in text how to solve these issues

Overall, significant amount of data is displayed in Figures.

Should show the standard deviation error bars. Revise also not all the needed parameters displayed there, Figures can be sharper, showing the detailed info with error bars.

All figures needed to be mentioned in text. Figures need to be done in Origin or better software. Legends not visible enough.

what was stable conditions in sharpening with error estimations +-?

Errors and error bars missing some places and add them elsewhere too.

Following a comprehensive analysis of data, the results were promising . Need to clarify sentences adding numerical results 

No essential novelty at different conditions can be drawn. As clearly seen also by Fig.

Intro part should be 1.5 pages.

The data that support the findings of this study should added with the ref as per format of references

Line 28-  „fuzzy  entrop (FE)“ – should be entropy

After: “An improved HMRDE is proposed to extract obvious difference features with various frequency ranges, which defeats the drawback that MRDE omits frequency change features. Literature have shown different environmental water problems to be solved for more economic way by modelling, which could be shown: https://doi.org/10.3390/w14020242, https://doi.org/10.3390/w13152136, https://doi.org/10.3390/w13141969,  https://doi.org/10.1016/j.scitotenv.2021.149133, https://doi.org/10.3390/w13030350, and afterwards there could be cited in newer publications regarding ORP control: https://doi.org/10.1089/ees.2018.0225 and electricity production combined with wastewater treatment. DOI 10.1007/s10532-020-09907-w

  • Gituku, E.W.; Kimotho, J.K.; Njiri, J.G. Cross-domain bearing fault diagnosis with refined composite multiscale fuzzy entropy and the self organizing fuzzy classifier. Eng. Rep. 20213, 12307. [Google Scholar] [CrossRef]
  • Songrong, L.; Wenxian, Y.; Youxin, L. Fault Diagnosis of a Rolling Bearing Based on Adaptive Sparest Narrow-Band Decomposition and RefinedComposite Multiscale Dispersion Entropy. Entropy 202022, 375. [Google Scholar]
  • Li, Y.; Gao, X.; Wang, L. Reverse Dispersion Entropy: A New Complexity Measure for Sensor Signal. Sensors 201919, 5023.
  • https://doi.org/10.3390/s22052046

Reviewer 2 Report

It would be very good to improve the quality of Fig.6

Reviewer 3 Report

In this study, the authors proposed an improved HMRDE to extract obvious difference features with various frequency ranges, which defeats the drawback that MRDE omits frequency change features. The paper is well written and easy to understand. To be published, the authors need to address the following issues. Therefore, I recommend that this paper be accepted for publication after revisions.

The authors should change the title of the paper.

The authors should enlarge the introduction section.

All figure labels and legends should be made more readable and should be in the same font.

The authors should state the originality of the paper at end of the introduction.

The authors should rewrite the conclusion.

The authors should explain the advantages and disadvantages of HMRDE clearly.

Round 2

Reviewer 1 Report

Thanks for thorough review. Novelty aspects can be better introduced as well as adding more fewer studies on relevant topic. Figures should not use different color codes, but unified codes could be helpful. The number of figures is to much for scientific paper, it could be diminished whenever possible. Just as advice not big recommendation, for future. Figure 6 is much smaller than other figures, sizes differences of figures need to be figured out. Checking latest literature could change manuscript better: https://doi.org/10.1016/j.scitotenv.2021.149133, https://doi.org/10.3390/w13030350 , https://doi.org/10.3390/s22052046, https://doi.org/10.3390/w13111522, https://doi.org/10.3390/w13141969, https://doi.org/10.3390/w14020242, https://doi.org/10.3390/w13152136

Figure 5. The frequency spectrum of time series under two health conditions- figure sections a) and b) need to be on Figure. Many empty space on figure not needed. Overall MS brings interesting and thorough results, which have meaning to the society. After some little revision the MS can be accepted.

Reviewer 3 Report

The authors have well addressed all my comments, I suggest this paper can be accepted for publication in the journal of entropy.